# Microprocessor-Controlled Prostheses for a Bilateral Transtibial Amputee with Gait Analysis and Satisfaction: A 1-Year Followup Case Report

**DOI:** 10.3390/ijerph19148279

**Published:** 2022-07-06

**Authors:** Jong Hu Jeon, Hee Seung Yang, Pyoung-hwa Choi, Eui Jin Ahn, Woo Sob Sim, Dong Young Ahn, Jin Yang Kim, Seul Bin Na Lee, Min Jo, Min Hee Cho

**Affiliations:** 1Veterans Health Service Medical Center, Department of Physical Medicine and Rehabilitation, Seoul 05368, Korea; w.hisstory@gmail.com (J.H.J.); ej6579@naver.com (E.J.A.); 2Veterans Health Service Medical Center, Center of Prosthetics and Orthotics, Seoul 05368, Korea; peace8422@naver.com (P.-h.C.); s2ellove@bohun.or.kr (W.S.S.); dyhhjh@bohun.or.kr (D.Y.A.); plcas1@naver.com (J.Y.K.); seulbinna@gmail.com (S.B.N.L.); min8606@bohun.or.kr (M.J.); mhe828813@gmail.com (M.H.C.)

**Keywords:** microprocessor-controlled prosthetic feet, prostheses, bilateral amputee, gait analysis, questionnaire

## Abstract

Bilateral amputees are disadvantaged as they lack healthy leg support. We present the rare case of a bilateral transtibial amputee (BTA), in which we compared the first South Korean-made microprocessor-controlled prosthesis (MPA) to a conventional prosthetic ankle (CPA) with gait analysis and a patient questionnaire for long-term outcomes. A 70-year-old man presented with bilateral transtibial amputations from injury. Assessments were performed after wearing an MPA for 1 month and 1 year with three-dimensional gait analysis. Satisfaction, mobility, and pain were evaluated using the Korean version of the Prostheses Evaluation Questionnaire (K-PEQ). The spatiotemporal parameters of both sides showed increased stability from the CPA to 1 month (mMPA) and 1 year (yMPA). We observed an increased single support time, decreased step width, and almost normal stance-swing time ratio. In kinematic parameters, the ankle range of motion (ROM) was bilaterally increased at mMPA and yMPA. Unfortunately, the MPA gait showed insufficient ankle plantarflexion during the terminal stance that failed to generate push-up power. As the MPA adaptation time increased, the symmetry ratio improved to a balanced value. The questionnaire-based investigations of satisfaction, mobility, and pain revealed excellent results. The MPA proved helpful for ankle mobility in the BTA, and the questionnaire showed good satisfaction and mobility in varied terrain.

## 1. Introduction

Limb amputation is a major change that constrains independent living and quality of life [1,2]. Corresponding with the development of medical technology and prolonged lifespan, loss of the lower extremities from chronic diseases is gradually increasing in frequency [3,4,5,6]. Proper selection of a prosthetic limb to replace the lost extremity is important for rehabilitative gait training [7,8]. Training enables the performance of activities of daily life [9]. It is a critical process that determines the degree of functional ability.

Historically, most lower extremity amputees mainly use artificial knees and feet that passively adjust the joint angle with elastic systems such as springs, hydraulics, and air cylinders. The passive ankle prosthesis is widely used, but it has only one joint resistance, making it impossible to control and offer active feedback while responding to slippery or inclined walking terrain [10,11]. Even when attempts are made to perform appropriate gait rehabilitation, the risk of falls is very high due to the lack of ability to change walking speed, high energy consumption, and low walking stability [11,12,13,14].

To overcome this limitation, the microprocessor-controlled prosthesis (MPA) has been used increasingly since its introduction in the 1990s. The MPA responds actively to gait patterns and the surrounding terrain. Its internal microprocessors control passive characteristics and active movements at both knee and ankle joints. Use of the MPA has increased confidence and safety, reduced cognitive loading, improved energy efficiency, and increased self-selected walking speeds [15,16,17,18,19].

Previously, there have been many papers comparing the recently developed MPA with conventional prosthetic feet (CPF) to lower extremity amputation patients. However, most of the prosthetics used as CPF are not used much now [16], and most of the studies were conducted on unilateral amputees [20].

Compared to unilateral amputees, bilateral amputees are disadvantaged when it comes to ambulation and gait training. They lack the support of a healthy leg. Nevertheless, it is believed that MPAs could also improve the gait for a bilateral transtibial amputee (BTA). There were cases where MPA was applied to bilateral transfemoral amputation, not transtibial level, and the followup period was short [8,21]. The evaluation after the long-term application of MPA in varied terrain can be a strong basis for practical use, but the subjective satisfaction of patients is also very important. Few well-designed studies have been conducted on this to date, despite its importance.

Here, we present the case of a BTA, comparing the first South Korea-made MPA to a conventional prosthesis according to gait analysis and patient satisfaction. Especially with a long-term followup of one year, this will be a meaningful comparison.

## 2. Materials and Methods

### 2.1. History

The participant (male, age 70 years, 168.7 cm, 62.4 kg) had injuries to both lower extremities from a landmine explosion in the Vietnam War in 1971, for which he had undergone bilateral transtibial amputation. On presentation, the right side amputation was at 9.9 cm and the left at 7.5 cm below the tibial plateau (Figure 1a,d). Both stump surfaces were cylindrical. The soft tissue shrinkage was intact and healed (Figure 1c). The left lower extremity had a knee flexion contracture of 10° after a distal femur malunion (Figure 1b). He had a functional K-level of 3.

He received initial prostheses in 1972, composed of a solid ankle cushion heel (SACH) foot and exoskeletal shank, a patellar tendon-bearing socket, and a fork-strap with a waist belt. From 1972 to 2014, the prosthesis was only used indoors and rarely when going out because of the energy expenditure and discomfort. In 2014, he began wearing the dynamic response feet, Vari-Flex^®^ (Össur EHF, Reykjavik, Iceland). This is a conventional prosthetic ankle (CPA), with a total-surface-bearing design that uses supracondylar pin-shuttle locking suspensions on silicon liners due to the short stump length.

The first South Korean MPA, the RoFT^®^ (Hugo dynamics, Daegu, Korea), was developed in 2017 (Figure 2). Since an MPA has sensors in the ankle joint and on the foot sole, it recognizes varied walking terrain and optimizes movement and force, increasing safety. In 2021, the patient was introduced to the MPA. Gait analysis was performed after an adaptation period of approximately one month. Followup continued for one year (Figure 3).

### 2.2. Methods

The patient’s gait was recorded with a three-dimensional motion analysis system^®^ (Motion Analysis Corp., Santa Rosa, CA, USA). Markers were attached to anatomical landmarks, and markers for the amputation site were attached by estimating the location on the MPA (Figure 1e,f) [22]. The system tracked the attached 19 markers (Qualysis AB, Goteborg, Sweden) with diameters of 12.5 mm using eight infrared cameras (Raptor-12HS^®^, Motion Analysis Corp., Rohnert Park, CA, USA), positioned around the treadmill. Raptor-12HS^®^ is considered the best of the motion capture cameras, operating up to 300 Hz at a full resolution of 12.5 Mpixel in a FOV of 64° × 50°. The marker trajectory data were recorded throughout all trials at 100 Hz and recorded in Cortex software^®^ (Motion Analysis Corp., Santa Rosa, CA, USA).

Relative and absolute joint positions were quantified with a tracked marker and calibrated using nonlinear transformation [23]. The record of the tracked markers was analyzed with OrthoTrak^®^ 6.4.4. (Motion Analysis Corp. Santa Rosa, CA, USA).

The ground reaction force (GRF) was collected and calculated through three Kistler 9260AAA^®^ (Kistler Corp., Amherst, NY, USA) platforms. The patient walked at a self-selected speed 15 times along a 10-m walkway. Sufficient rest breaks were taken between each of the 15 “laps.” The initial assessment was completed using the CPAs and the second and third assessments using MPAs after 1 month and 1 year of wear. From the 15 analyses, we chose the most informative components of the evaluation to generate spatiotemporal, kinematic, and kinetic parameters. Symmetry ratios for step lengths and intra-cycle gait phases were calculated by dividing the value of the right foot variable by that of the left foot [24].

The survey was prepared based on the Korean version of the Prosthesis Evaluation Questionnaire (K-PEQ) for the respective compatibility of MPA and CPA [25]. From the eight K-PEQ categories, we selected the four categories of satisfaction, physical sensation, mobility, and unexpected situation occurrence. The referenced questionnaires were divided into three categories: satisfaction, physical sensation, and mobility. Each question was scored on a scale of 1 to 10 using the Likert scale. For questions 1–17 and 22–33, higher scores meant higher satisfaction; for questions 18–21, lower scores meant higher satisfaction. Moreover, the participant was asked about the pros and cons of MPA with open-ended questions.

## 3. Results

### 3.1. Spatiotemporal Parameters

At the 1-month MPA (mMPA) and 1-year MPA (yMPA) assessments, both sides showed increased single support time and decreased step width. Stance-swing time ratios were approximately 7:3 for CPA bilaterally, and gradually improved to approximately 6:4 for both mMPA and yMPA, which was close to normal. This was more pronounced on the left side (Table 1). The symmetry ratio approached more perfect balance at mMPA and yMPA than CPA, except for step length and ankle plantarflexion moment.

### 3.2. Kinematic Parameters

The ankle plantarflexion peak angle at loading response increased for both mMPA and yMPA and was far closer to normal than with CPA (Figure 4A(c),B(c) solid arrow). The ankle dorsiflexion peak angle in the late stance increased more for yMPA than for mMPA (Figure 4A(c),B(c) white arrow). For this reason, the total ankle ROM was the largest for yMPA, followed by those for mMPA and CPA, in that order.

### 3.3. Kinetic Parameters

At the mMPA assessment, both ankle plantarflexion peak moments in the terminal stance decreased compared to those in the CPA (Figure 4A(f),B(f) solid arrow). Thus, ankle power did not increase (Figure 4A(i),B(i) solid arrow). However, the yMPA measurements increased in moment and power compared to those at mMPA and were similar to those with the CPA (Figure 4A(i),B(i) solid arrow).

### 3.4. Questionnaire

The MPA was better in balance, appearance, durability, and compatibility (Figure 5 (Q6, 9, 12, 13)) from a mechanical point of view not only for skin problems (Figure 5 (Q14–17)) but also for related pain and was rated better than the CPA, where pain was felt at the stump site and lower back (Figure 5 (Q18–21)). In addition, the gait with the MPA scored high in various walking environments (Figure 5 (Q22–33)). No difference was noted on the stairs or downhill in mMPA, but the yMPA showed high improvement.

For the open-ended questions, the participant wanted the battery to last longer and felt uncomfortable with some noise. However, the weight which we were concerned with did not matter much. Rather, he was satisfied that daily life had become much better than before and waswilling to continue wearing the MPA.

## 4. Discussion

The BTA usually encounters more disadvantages than unilateral amputees. According to Su PF et al. [26], compared to unilateral amputees, patients with BTA walk at slower speeds and lower cadences. They have reduced ankle dorsiflexion and knee flexion angle in peak-to-peak stance phase (loading response), reduced ankle plantar flexor peak moment, and reduced ankle power in late stance. Various prosthetic technologies have been introduced to support the absence of the ankle plantar flexor muscles and moment in particular. The newly-produced RoFT^®^ MPA from South Korea is one of these.

As in the BTA patient described above, walking velocity and cadence were decreased for mMPA and yMPA when compared to a normal gait. The step length also decreased for mMPA, but increased somewhat on the right for yMPA. This is expected because of the weight. The MPA weighed 1750 g, which is more than 10 times heavier than the CPA (130 g). Thus the stress on the socket-residual limb interface increased. Gitter et al. assessed the addition of 1.34 kg of mass [27]. In that case, an increase of 5.4 J was required to accelerate the heavier prosthetic limb into the swing phase. The weight of the prosthetic leg is also important in terms of patient preference and gait efficiency [24]. In our yMPA assessment, there was an increase in step length. This seems to have improved with MPA adaptation. The increasing subjective satisfaction rating about weight also confirmed patient adaptation. Nevertheless, the weight disadvantage should be addressed in the future. This could improve the yield for various parameters.

For patients with BTA, an increase in step width usually occurs due to poor walking stability [26,28]. In this case, however, the step width measured for mMPA and yMPA decreased significantly, which meant increasing stability. Further, the answers on the mobility questions about stability on various terrain support this result.

When a prosthetic foot is worn, heel contact time is delayed, and an early heel rise occurs. This shortens the ground contact period, which in turn delays the toe-off of the opposite. Accordingly, the loading response phase (LRP) occupies close to 20% of the gait cycle with the CPA and 12% in normal patients without a prosthesis [29]. In contrast, the LRP for the mMPA and yMPA were close to normal. This is because the heel-only period was shortened. Ankle movement occurred before the shank moved forward due to increased plantarflexion within the initial stage of LRP on both sides. This made the time of the foot-to-ground contact shorter than with the CPA.

Under normal conditions, a plantarflexion moment would be generated, which corresponds to the dorsiflexion [29]. However, the right ankle plantarflexion moment during LRP was not sufficient with the CPA or MPA, although it stored energy during dorsiflexion. Neither the CPA nor the MPA generated as much plantarflexion power. Additionally, in unilateral amputation, the unaffected side provides sufficient stability, and there is sufficient time for toe-off to occur sequentially after heel rise. However, in BTA, the ankle moment and power are reduced as described above, because the time when MPA provides power for plantarflexion is shortened. Therefore, the shank movement is reduced, and the swing phase knee flexion angle is decreased, so that dorsiflexion is slightly extended. According to the results of the yMPA, which were higher than that of the mMPA, it seems encouraging to expect that more stable support time can secured in the future.

Compared to a normal gait, vertical movement is usually decreased in BTA [29]. Our patient also did not have much up and down movement in the center of mass. However, as the MPA adaptation time increased, the symmetry ratio improved to a balanced value (Table 1). In particular, the ankle plantarflexion power symmetry ratio showed significant improvement. The questionnaire answers regarding balance also yielded high marks for MPA, supporting this idea (Figure 5 (Q6)).

Today, various lower extremity prosthetic devices exist. Each device has a different design and function [30]. One of the relatively new and emerging types of prosthetics is one controlled by a microprocessor. The RoFT^®^ MPA is one such prosthesis, with the ability to attain input information for the various environmental factors and make a strategic output of the device. Although RoFT^®^ has the advantages described above as an MPA, it also needs to be compared with other MPAs. There are MPAs that have tried to reduce the weight, and there are MPAs with different designs, so comparison with other MPAs is important [31,32]. This study was meaningful in that MPA was applied to a BTA and followed for a long time; however, the study only followed on person. In a further study, it would be good to evaluate more patients and performance on a sloped terrain.

## 5. Conclusions

The new RoFT^®^ MPA enabled our patient to walk in a more physiologic manner due to the increased single support time, symmetry ratio, and improvements in ankle kinematics. However, the decreased plantarflexion power produced less physiologic kinematics and kinetics, which was negative. The prosthetic weight and operating noise can be improved. Nevertheless, after 1 year of followup, significant increases in moment, power, and balance were observed. Because outdoor terrain is more diverse than indoors, gait training and living with various conditions are important. Our patient reported improved satisfaction when walking in varied terrain with more efficiency. This is encouraging for the MPA. These analyses and positive feedback should be helpful in continued MPA development and safe gait training in varied terrain for patients with BTA.

## Figures and Tables

**Figure 1 ijerph-19-08279-f001:**
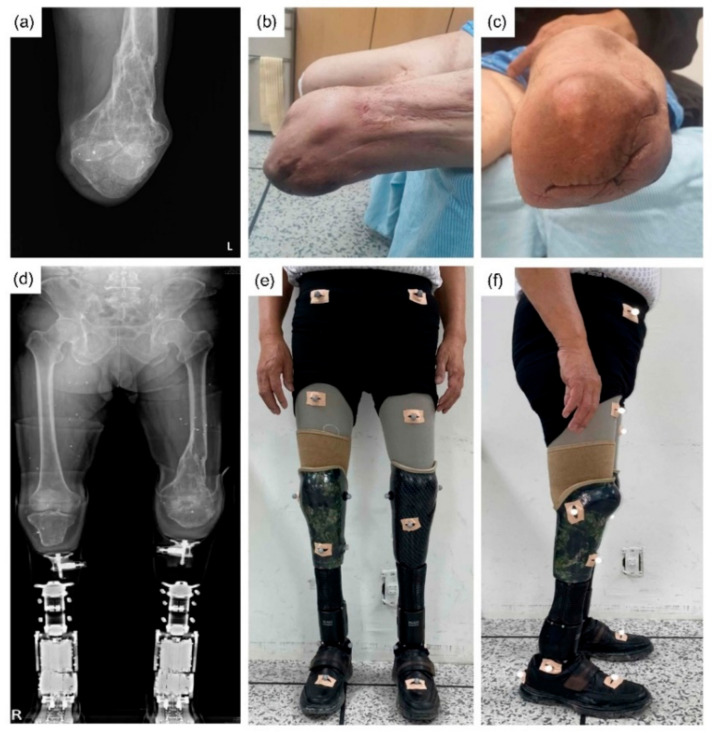
(**a**) X-ray of left knee flexion contracture due to distal femur malunion. (**b**) Flexion contracture of 10° of the left leg. (**c**) Cylindrical left stump with intact soft tissue shrinkage in healing state. (**d**) Lower extremities X-ray view with RoFT^®^. (**e**,**f**) The front and side view of participant wearing RoFT^®^. 19 sensors attached with the Helen-Hayes method to the anterior superior iliac spines, sacrum, thighs (midpoint between the greater trochanter and the lateral femoral head), knee joint, tibial tuberosity, ankle medial and lateral malleolus, heel, and second metatarsal bone. With bilateral prostheses, it is difficult to attach sensors in the correct anatomical position.

**Figure 2 ijerph-19-08279-f002:**
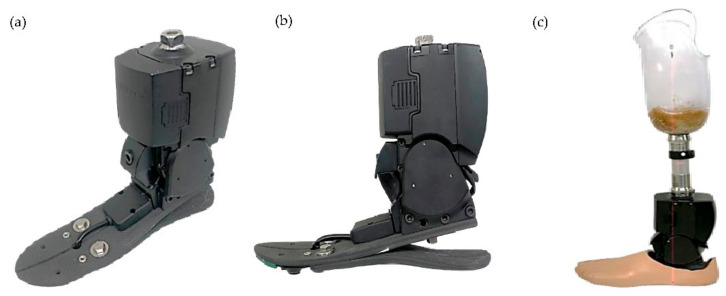
(**a**) Oblique and (**b**) side view of exterior of the RoFT^®^ (Hugo dynamics, Daegu, South Korea). (**c**) RoFT^®^ with socket. The newly produced MPA in South Korea, RoFT^®^, works through a motor-driven method and weighs about 1750 g. It uses a battery with a usage time of 4 h after being fully charged, but it can be replaced detachably. Its maximal dorsiflexion angle is 20° and maximal plantarflexion angle is 15°.

**Figure 3 ijerph-19-08279-f003:**
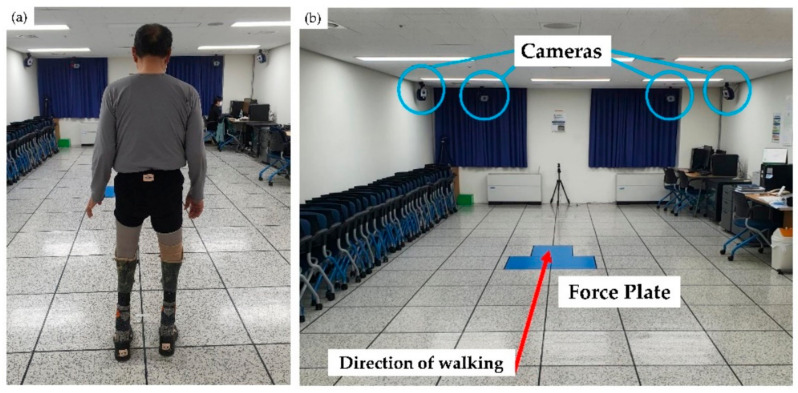
Experimental setup in the laboratory. (**a**) The participant standing in front of the force plate with 19 markers placed at landmarks. (**b**) The eight infrared cameras were mounted in front and back of the walkway.

**Figure 4 ijerph-19-08279-f004:**
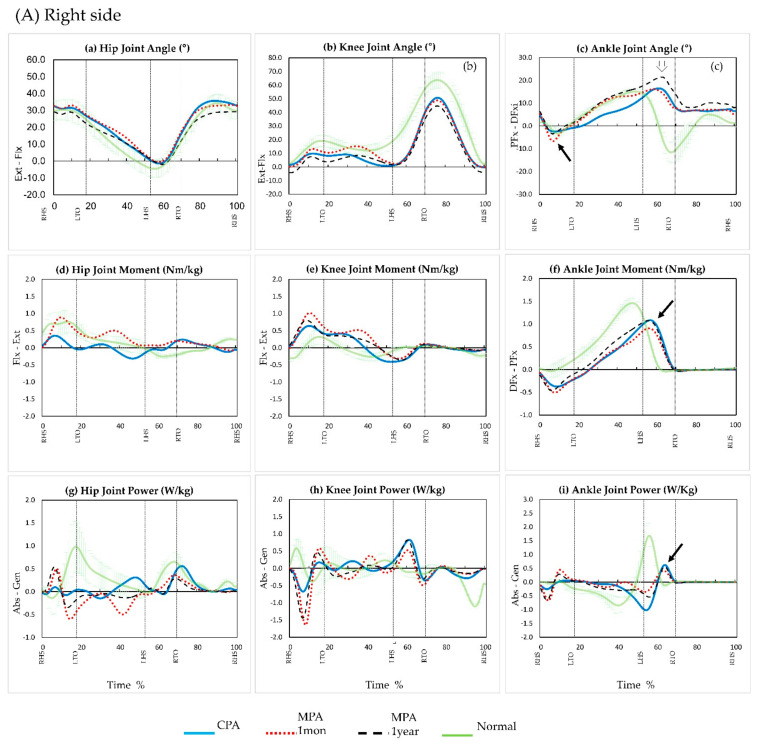
Graphs illustrating sagittal plane motions, moments, and power of the hip (**a**,**d**,**g**), knee (**b**,**e**,**h**), and ankle (**c**,**f**,**i**) across the gait cycle (0 to 100%) during walking. The green solid line represents group mean data for normal non-amputee individuals. The blue line represents the group mean data for the conventional prosthetic ankle. The red dashed line represents the group mean data for a microprocessor-controlled prosthetic knee after 1 month. The black dashed line represents the group mean data for a microprocessor-controlled prosthetic knee after 1 year. Abbreviations: Flexion (Flx), Extension (Ext), Plantarflexion (PFx), Dorsiflexion (DFx), Absorption (Abs), Generation (Gen), Right heel strike (RHS), Left heel strike (LHS), Right toe off (RTO), Left toe off (LTO), Right heel stroke (RHS), Left heel stroke (LHS).

**Figure 5 ijerph-19-08279-f005:**
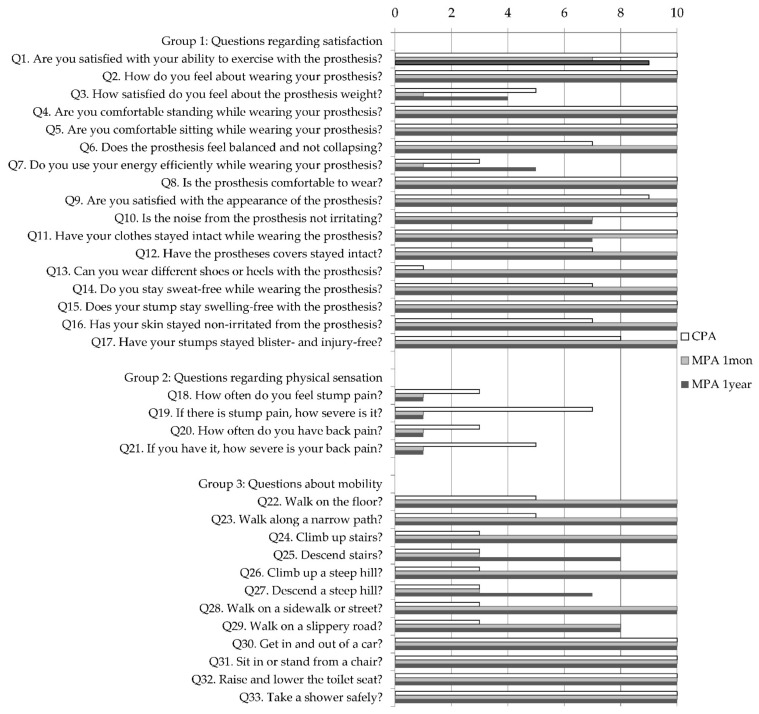
Questionnaire regarding satisfaction and mobility after using CPA and MPA for 1 month and 1 year. It is referenced from the Korean version of the Prosthesis Evaluation Questionnaire (K-PEQ) and divided into three categories: satisfaction, physical sensation, and mobility. Each is scored 1–10 points; for Q1–17 and Q22–33 higher scores indicate higher satisfaction, and for Q18–21, lower scores indicate higher satisfaction.

**Table 1 ijerph-19-08279-t001:** Comparison of gait analysis between CPA, 1-month MPA, and 1-year MPA.

	CPA		MPA (1 Mon)		MPA (1 Year)	
	Right (*n* = 8)	Left (*n* = 8)	Right (*n* = 11)	Left (*n* = 11)	Right (*n* = 9)	Left (*n* = 9)
**Spatiotemporal parameters**						
Forward velocity (cm/s)	63.27 ± 3.62	64.36 ± 2.60	57.56 ± 4.40	57.05 ± 4.77	57.95 ± 6.74	58.97 ± 7.00
Cadence (step/min)	85.07 ± 1.92	85.83 ± 1.80	78.79 ± 3.18	79.11 ± 3.55	81.19 ± 3.14	82.13 ± 4.22
Step length (cm)	45.72 ± 3.17	43.91 ± 3.30	41.71 ± 3.61	44.96 ± 2.56	44.21 ± 5.04	41.25 ± 3.42
Stride length (cm)	89.96 ± 6.21	90.37 ± 4.7	86.09 ± 5.75	86.71 ± 6.98	84.71 ± 7.64	86.03 ± 7.52
Step width (cm)	28.31 ± 1.03		28.08 ± 0.99		26.93 ± 1.64	
Stance time (%cycle)	70.58 ± 1.51	68.13 ± 2.45	69.95 ± 1.80	62.09 ± 1.76	70 ± 1.87	63.77 ± 1.78
Swing time (%cycle)	29.42 ± 1.51	31.87 ± 2.45	30.05 ± 1.80	37.91 ± 1.76	30 ± 1.87	36.23 ± 1.78
Initial double support time (%cycle)	21.43 ± 1.45	16.81 ± 0.91	16.83 ± 1.51	15.26 ± 1.59	16.78 ± 1.64	14.75 ± 1.24
Total double support time (%cycle)	38.71 ± 2.39		32.04 ± 3.42		33.77 ± 2.53	
Single support time (%cycle)	31.87 ± 2.45	29.42 ± 1.51	37.91 ± 1.76	30.05 ± 1.80	36.23 ± 1.78	30 ± 1.87
Single:Double support time ratio	31.87:38.71	29.42:38.71	37.91:32.04	30.05:32.04	36.23:33.77	30.00:33.77
**Kinematic parameters**						
Maximal hip flexion angle (°)	35.76 ± 1.08	26.50 ± 0.87	34.03 ± 1.03	28.94 ± 1.04	30.22 ± 2.16	23.86 ± 1.39
Knee flexion angle in stance phase (°)	10.82 ± 2.50	-	15.62 ± 4.06	-	9.62 ± 4.46	-
Ankle plantarflexion peak angle at loading response (°)	−2.52 ± 0.56	0.48 ± 0.40	−6.97 ± 0.93	−2.41 ± 0.49	−3.96 ± 0.94	−4.61 ± 0.62
Ankle dorsiflexion peak angle in late stance phase (°)	16.67 ± 0.92	17.63 ± 0.69	16.33 ± 0.89	17.45 ± 1.58	21.84 ± 1.03	19.80 ± 1.28
Ankle ROM in stance phase (°)	19.19 ± 0.93	17.15 ± 0.72	23.30 ± 1.11	19.86 ± 1.51	25.79 ± 1.73	24.41 ± 1.69
**Kinetic parameters**						
Peak ankle plantarflexion moment in late stance phase (Nm/kg)	1.09 ± 0.12	0.90 ± 0.08	0.91 ± 0.09	0.58 ± 0.17	1.08 ± 0.07	0.78 ± 0.13
Peak positive ankle power (W/kg)	0.67 ± 0.12	0.21 ± 0.05	0.45 ± 0.10	0.25 ± 0.21	0.60 ± 0.16	0.47 ± 0.23
**Symmetry ratios** ^1^						
Step length ratio	1.04		0.93		1.07	
Early stance ratio	1.27		1.10		1.14	
Late stance ratio	0.79		0.91		0.89	
Ankle ROM ratio	1.12		1.17		1.06	
Hip flexion ratio	1.35		1.18		1.27	
Ankle plantarflexion moment	1.21		1.56		1.40	
Ankle plantarflexion power	3.21		1.83		1.27	

Values are mean ± standard deviation. ^1^ Symmetry ratios for step lengths and intra-cycle gait phases were calculated by dividing the value of the variable of the right foot by that of the left foot. A value of 1 means perfect bilateral balance; a value >1 means right-biased.

## Data Availability

Data will be made available on reasonable request from the corresponding author.

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
