# Peer review of "Microprocessor-Controlled Prostheses for a Bilateral Transtibial Amputee with Gait Analysis and Satisfaction: A 1-Year Followup Case Report"

_ijerph, 2022, doi:10.3390/ijerph19148279_

Round 1

Reviewer 1 Report

I would like to congratulate the authors for the work they have done. Although case reports do not provide a high level of evidence, the authors described adequately an interesting clinical case. 

The few comments I would like to sign are:

1) To include a reference for the reporting of clinical cases (i.e., Riley DS, Barber MS, Kienle GS, et al. CARE guidelines for case reports: explanation and elaboration document. J Clin Epidemiol. 2017;89:218-235. doi:10.1016/j.jclinepi.2017.04.026)  

2) To include a "take-away" message from this case report at the end of discussion. 

3) To share also the patient's point-of-view

Reviewer 2 Report

The article is interesting and valuable. The artucle has a correct IMRAD structure. The introduction section should be edited in such a method that the article takes on a scientific character. In the results section, the article is written with the wrong ratio.

1. What algorithm is used to determine the trajectory based on the movement of markers in time?

2. What is the accuracy of the optical system?

Major:

Authors should articulate the research gap in the introduction section.

The number of literature references should amount to about 25 items.

The resolution of Figure 2 should be increased.

Only one person was subjected to examination.

You should enter the diameter of the markers in the section Matherials and Methods

What was the marker sampling frequency?

Is the value Forward velocity in meters per second in the Table 1?

Minor:

You should put some text below Figure 3.

You should describe the statistical apparatus used.

Reviewer 3 Report

The paper discusses about the effect of a prosthesis in one month to one year comparison. There are some questions that is need to be answered

1. What is MPA the roft hugo dynamics? can you put more details and picture/schematic/block diagram of it? Providing this information can help the reader to understand your intervention to the patient.

2. Experiment set up can also be shown using picture. 

3.Figure 2 should be separated so it can be seen in a bigger and clear picture. Also, the figure doesnt have legend for the colour.

4. references are not enough, I suggest you to discuss other prosthesis or orthosis with different control method and also subject for interview. Also most of the references are not new.

Round 2

Reviewer 2 Report

The authors responded to comments with appropriate scientific logic.From the point of view of the correct presentation of the experiment, the article gained importance. The work is going in the right direction, but the number of participants in the experiment should be increased. This is the biggest drawback of the article.